# A Qualitative Study of Irish Dairy Farmer Values Relating to Sustainable Grass-Based Production Practices Using the Concept of 'Good Farming'

Orla Kathleen Shortall 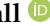

James Hutton Institute, Aberdeen AB15 8QH, UK; orla.shortall@hutton.ac.uk

**Abstract:** Ireland's grass-based dairy system is relatively unique in industrialised countries in its focus on producing milk from grazed grass rather than increasing yields through non-forage feed. The environmental benefits of a grass-based dairy system have been promoted within Ireland and abroad. However, the means by which grass is produced is important. There have been environmental concerns about water pollution from nutrient leaching and increasing greenhouse gas emissions from the increased number of cows and higher fertiliser application in the Irish dairy sector. This paper uses qualitative interviews with Irish dairy farmers to assess: (1) how can we understand Irish farmer attitudes towards the grass-based system within a 'good farmer' theoretical framework? (2) How do concepts of extensive and intensive production fit with good farming norms within the grass-based system? (3) How could cultivation of multispecies swards, including legumes, fit with existing notions of good farming? The research finds that there had been a concerted efforts by researchers, advisory bodies and other actors to foster a definition of good farming to mean good grass management. This definition of good farming excluded the use of feed inputs over a certain level to increase yields but included the use of fertiliser to maximise grass production. There is scope to change the definition of good farming within the industry to include minimal use of fertiliser, for instance through the cultivation of multispecies swards including legumes and the skills and knowledge this involves. In terms of policy implications, the paper identified three strategies for government and industry to facilitate a definition of good farming which involves low fertiliser use: emphasising the cost-saving aspect of reducing fertiliser; identifying visible symbols of 'good farming' using multispecies swards; and co-producing the definition of good farming with a diverse range of stakeholders including farmers.

**Keywords:** dairy; Ireland; grass; grazing; qualitative; greenhouse gas emissions; nutrient pollution; good farming

## 1. Introduction

The role of grass and grazing has decreased in importance in dairy sectors in many industrialised countries, with cows fed concentrate and grain and spending more time indoors to increase yields [1]. Ireland is something of an exception to this trend with 95–100% of dairy farms grazing [1]. Some 90% of Irish farms calve in spring, meaning milk can be produced from seasonal grass growth, and 90% of Irish dairy produce is exported in processed form which requires a high fat and protein content; this circumstance is suited to a grass-based system.

Researchers have argued that there is a need for better understanding of farmer attitudes towards grazing and measures that facilitate the uptake of grazing practices because of the environmental benefits of grazing [1]. Previous research has used surveys to explore attitudes towards grazing with farmers in Germany [2] and Denmark [3] and the adoption by Irish farmers of particular grass management technologies such as grass measurement [4,5], paddock grazing [6], grass budgeting [7] and the use of a spring

rotational planner [8]. This paper uses a theoretical framework grounded in research on 'good farming' [9] to add further depth to research on farmer attitudes toward grass-based systems. In addition, how grassland is managed affects nutrient cycling and environmental footprint [10,11]. This paper also explores farmer views of 'extensive' and 'intensive' grass management and how a potential lower-input grass-based practice of using multispecies swards fits with existing good farming ideals, where 'intensive' is taken to broadly mean agricultural production that uses more inputs and 'extensive' broadly involves fewer inputs [12].

As the next section describes in more detail, the Irish government and industry promote the environmental sustainability of the Irish dairy sector because it is grass based [13], but the recent expansion of the sector following the removal of EU milk quotas in 2015 has resulted in negative environmental outcomes [14]. Successfully reducing greenhouse gas emissions and nutrient leaching from the dairy sector will involve changes in practices by dairy farmers. This paper focuses on farmer attitudes and practices relating to the current grass-based system in Ireland and grass-management practices that have the potential to reduce greenhouse gas emissions and nutrient pollution, in particular the use of multispecies swards. A multispecies sward is one that includes herb and legume species as well as grasses. Legumes such as clover and sainfoin have the ability to fix nitrogen in the soil and thereby reduce the amount of fertiliser required, which reduces the resulting nitrous oxide omissions [15], and herbs such as chicory and plantain have deep roots to access minerals deeper in the soil and are more resistant to drought [16,17]. The use of perennial rye grass dominates the Irish dairy sector, making up 95% of grass seed sales [18]. In the 20th century, research and development focused on perennial rye grass because of its early spring growth, high digestibility and palatability for cattle, high dry matter content and ability to regrow after defoliation [19]. Fertiliser use to enhance the growth of perennial rye grass was incentivised by government subsidies beginning in the 1950s [18]. Fertiliser use led to a reduction in grassland diversity as most native grass species and herbs cannot compete with perennial rye grass when fertiliser is applied [18].

Social sciences researchers have examined how cultural factors influence the behaviour of land managers [20]. The concept of the 'good farmer' was developed to analyse the kinds of practices, identities and skills that are valued in different farming communities [9]. There are two main theoretical underpinnings to the good farmer concept: Pierre Bourdieu's theory about economic, social and cultural capital and a symbolic interactionist perspective which looks at how people create and maintain meaning and identity within the social world [9]. According to Bourdieu's conceptualisation, capital is accumulated labour in material or embodied form [21]. Economic capital is material assets which can be readily converted into money; cultural capital consists of embodied dispositions such as skills, cultural material goods and institutionalised goods such as qualifications; and social capital is the potential or actual resources that can be accessed through networks and 'connections' [21]. Being a good farmer means accumulating different kinds of capital within common 'rules of the game' [22]. This paper will use this framework of good farming as striving for different kinds of capital, which will be referred to in the results section. It is also worth noting that terminology used in the good farming literature varies: terminology of identities and different types of capital is widely used. This paper will also use the term 'norm', which is taken to mean an internalised value that is socially reinforced [23].

The early good farming literature consistently found that high production is a sign of good farming [24–26]. The good farming concept originally showed how valuing productivity meant that farmers were resistant to change towards environmental initiatives which might have reduced production [24,26]. Good farming symbols such as tidy fields [27] or healthy looking livestock [28] and identities are shaped by economic conditions [22,26]; technological development [9]; the influence of advisers [9,28,29]; and government policy [30–32] and are reinforced through 'roadside farming' where farmers look at their neighbours' fields for symbols of good farming [24].

Policies incentivising production increases and technological innovation led to substantial changes in production methods in 20th-century livestock production, including breeding for higher yields, greater stocking density, more housing of animals to allow for greater control and increased use of purchased feed rather than grazing [33,34]. Productivist agricultural policy was replaced to some extent in the EU, or at least augmented since the 1990s, by the policy trajectory of multifunctionality or post-productivism [35]. This involves encouraging and supporting farmers to produce goods on farms other than commodities, such as diversification and environmental schemes, and supporting rural development [35] as a way around the ills caused by productivist 'monofunctional' agriculture [36,37].

Recent good farming literature has demonstrated how norms can and do change in response to changing market and policy landscapes and has shown a move away from productivist good farming identities. Because of reductions in production subsidies, farmers have in some cases responded to a difficult economic environment by reducing inputs and yields to reduce costs [22,38,39]. Environmental stewardship has also been documented as a good farming norm [30–32,38–41], although there may be a ceiling on farmers' endorsement of environmental measures where they conflict with profitability [22,31,39]. Productivist norms are still relevant as farmers are juggling different identities in the face of changing rules of the game [27,40,42]. While there is a wider debate within research on agriculture about the economic and environmental benefits of grazing [1], the good farming literature has not considered the role of grazing in livestock production systems. This paper addresses this gap within the Irish context by asking to what extent grass-based dairy farming is defined by farmer interviewees as good farming. The paper aims to contribute to debates within Ireland and in the literature on grazing more widely about how best to meet environmental objectives in dairy farming by analysing whether intensive or extensive grass production methods are considered to be good farming.

This paper uses qualitative interviews with Irish dairy farmers to assess (1) how can we understand Irish farmer attitudes towards the grass-based system within a 'good farmer' theoretical framework? (2) How do concepts of extensive and intensive production fit with good farming norms within the grass-based system? (3) How could cultivation of multispecies swards fit with existing notions of good farming? The analysis relates to the Irish dairy sector but is relevant to other contexts. The first question is in line with calls for more understanding of why farmers graze, in order to facilitate uptake of grazing practices, which are declining in many countries [1]. The last two questions consider the challenge of promoting sustainable grazing practices, which are relevant in contexts where the dairy industry is predominantly grass-based or where grass is used in addition to significant amounts of non-forage feeds.

## 2. Background

This section provides background on the Irish dairy sector to help the reader understand the origins of the farmer interviewees' views on good farming presented in the results section. It explains why the Irish dairy sector has a unique focus on grass and the current environmental challenges the sector faces. The Irish dairy sector has recently undergone a period of significant expansion within the grass-based system following the removal of European Union milk quotas in 2015 [43]. Milk quotas stalled farm expansion and kept Irish dairy farms smaller than was considered economically optimal [44]. In the lead up to the removal of milk quotas, the Irish government set an ambitious target for the sector to increase milk production by 50% by 2020 [13]. This expansion was intended to happen through a continuation of the grass-based system: an increase in grass production and cow numbers [13]. Researchers at the Irish national food and research authority Teagasc argued that a system based on maximising milk from forage is the more economically rational [45,46]. A report written by two dairy industry stakeholders in 2015 cautioned that Irish farmers should keep to the grass-based system after milk quotas were removed and not increase yields through increasing purchased feed: 'The greatest danger to realising this [grass-based] potential is that farmers will drift away from grazed grass as the

foundation for low-cost, profitable milk production and sustainable, profitable farm family incomes'. [45] (p. 7).

Ireland has a unique extension landscape with a single body, Teagasc, carrying out research, education and extension, alongside a number of private agricultural advisors [47,48], compared with the rest of Europe, where farm advisory services are characterized by increasing diversity and privatization [49]. Research and extensive at Teagasc have for decades promoted the merits of the grass-based system [50]. Teagasc developed a dairy manual which set out the principles of the grass-based system [51]. The software Pasture-Base Ireland was developed to facilitate farmers' grass management decisions, allowing them to benchmark themselves in a national database of grass production [52]. A Dairy Efficiency Programme run by Teagasc from 2010 to 2012 encouraged adoption of best practices in cow breeding, grass management and profitability [53]. Teagasc has an extensive knowledge exchange network of discussion groups used as means of sharing knowledge among farmers [54]. Despite these measures, researchers state that grass production varies widely across dairy farms and that further increases in production are needed, with many dairy farming only achieving 50–60% of what they could produce [46].

Targets for increasing milk production by 50% in 5 years were largely achieved with a 40% increase in milk production between 2014 and 2019 [45]. There were concerns from environmental groups when expansion targets were proposed that an expansion of the national dairy herd would conflict with environmental targets in relation to greenhouse gas emission, biodiversity, ammonia emissions and water quality [55]. The 2010 government Food Harvest 2020 report [13] and its successor, FoodWise 2025 [56], both set out commitments to economic and environmental sustainability. Food Harvest 2020 described Ireland's grass-based livestock production system as inherently environmentally friendly: 'Ireland's extensive, low-input grass-based production systems are the foundation of its green credentials [...]' [2] (p. 5).

The Irish dairy system is seen as having lower greenhouse gas emissions per unit of produce than other countries: a European report showed Irish milk to have the lowest greenhouse gas (GHG) emission footprint in the EU [57]. Dairy farming is a source of a number of greenhouse gases: methane from enteric fermentation produced by the cow; carbon dioxide for embedded fossils fuels in feed, machinery use and loss of carbon from soils; and nitrous oxide from fertiliser and manure [58]. The carbon sequestered in grassland soils is seen as a factor making the Irish system more environmentally friendly than systems which buy in non-forage feeds, where soils tend to store less carbon or to emit carbon [57], although the topic of grassland carbon cycles is very complex, with claims that grassland may not sequester or may emit carbon depending on management [10].

Indeed, after continued dairy expansion, the Environmental Protection Agency made a bleak assessment of the environmental situation in 2019 in a response to a consultation for the agri-food strategy 2030. A report stated that the FoodWise 2025 strategy brought about intensification in production but not environmental protection. Evidence suggests that the intensification of farming has resulted directly in a deterioration in water quality, greenhouse gas emissions, biodiversity loss and ammonia emissions [14].

A report by the Department of Agriculture Food and Marine states that though emissions per unit of output decreased between 2015 and 2017, total greenhouse gas emissions from agriculture have increased, primarily because of the expansion of the dairy sector [59]. Ireland did not meet statutory 2020 climate change targets [60]. In relation to the expansion of the dairy herd, increased emissions have come from an expansion in cow numbers and use of inputs. Ireland has a derogation from the EU requirements for a maximum of 170 kg of livestock manure nitrogen per hectare to allow some farms 250 kg of livestock manure nitrogen per hectare. The number of farmers availing themselves of this derogation increased by 34% between 2014 and 2018 [59]. Fertiliser sales increased in 2017 and 2018, over 50% of which is used in the dairy sector [59]. Nitrogen fertiliser sales are projected to increase between 2020 and 2030 [59].

The Department of Agriculture, Food and the Marine set a target reduction of 50% of nitrous oxide emissions from fertiliser use by 2030 and a reduction of fertiliser use in Irish agriculture from its peak of 408,000 tonnes in 2018 to 350,000 tonnes by 2025 [61]. The Food Vision 2030 report commissioned by the Department for Agriculture, Food and the Marine set out a target of a climate-neutral agricultural system by 2050 with verifiable progress by 2030 [62]. The Ag Climatise strategy sets out grass management measures intended to reduce nitrous oxide emissions and water quality problems from the dairy sector including a requirement for clover in all grass reseeding by 2022 and a consideration of the use of legume crops [61]. There is also a government scheme that helps farmers with the costs of establishing multispecies swards [63]. A draft of the 2021 agri-food strategy received criticism for not being ambitious enough and for being a continuation of a model of agricultural intensification. The representative of the environmental group Environmental Pillar withdrew from the Strategy Committee prior to publication [64].

The research questions in this paper focus on the potential to meet environmental objectives for the Irish dairy sector through different grass management practices.

## 3. Materials and Methods

The research is based on qualitative interviews with dairy farmers in Ireland. Interviews, a qualitative research method, involves asking someone in-depth questions about what they do and what they believe [65]. The aim is to obtain detailed information on the interviewee's experiences and views on a particular topic. Qualitative interviewing involves carefully selecting a relatively small number of participants whose experiences are relevant to the research questions. The aim is not to generalise to a larger group of people, e.g., 'all dairy farmers think or do x', but to look in detail at the reasons why people do what they do and draw conclusions based on their circumstances [66]. Qualitative interviews were chosen because they were considered most appropriate for answering the wider project research questions about structural and cultural factors hindering or fostering change towards meeting policy objectives in the Irish dairy sector.

Ethical approval for interviews was obtained from the [removed for peer review] research ethics committee. Farmer interviewees were recruited through participation in a survey which was also carried out as part of the research project and which was disseminated between August 2018 and February 2019. In the survey, participants were asked if they were willing to take part in a follow-up interview. Out of 396 respondents to the survey, 92 indicated they were willing to be interviewed. Purposive sampling was used to interview people in a range of locations with different production systems and views [67]. The interviews were carried out in December 2019 and January 2020. All but one interview were carried out in person, so the aim of interviewing farmers in different locations was balanced with the logistical constraints of travelling to interviews within a given time frame. In total, 20 farmers were interviewed: 4 in the northeast, 2 in the midlands and 14 in the southwest of Ireland. The southwest was chosen as the location of the majority of interviews because this area has the highest concentration of dairy farmers in the country [68]. The number of interviews carried out was influenced by data saturation: the point at which no new information is being generated from additional interviews [69]. Notes were made after each interview to record the main findings to assess data saturation. The figure of 20 interviews is in keeping with the number of interviews normally carried out within qualitative research: for instance, McDonald et al. [7] carried out 8 narrative interviews with new entrants in addition to a survey; Regan et al. [4] carried out 21 interviews with farmers exploring their decisions whether or not to measure grass; and Kessler et al. [42] carried out 17 interviews with beef farmers in Canada on good farming values. All but one interviewee were male, as there were few female respondents to the survey. While the farmers interviewed are interested, motivated, engaged farmers because those are the people who are likely to fill in a survey and agree to be interviewed, their views speak to the wider culture in the industry as a whole. The interviews were semi-structured; there was an interview guide which was applied in a flexible way as farmers were asked

follow-up questions to elaborate on particular points. Farmer interviewees were asked to describe their farms, their views on the expansion of the dairy industry, challenges facing the dairy industry, views on grass-based and higher-feed-input systems and specifically about multispecies swards. Interviews lasted an average of 69 min, with the shortest being 36 min and the longest 124 min. Interviews were recorded and stored in a secure folder that was only accessible to the researcher. Interviews were sent to a third party for transcription. More details of the farmer interviewees' demographic details are shown in Table 1. The amount of concentrate farmers fed to cows is included because it indicates the extent to which they follow mainstream industry advice, from, for instance, the advisory body Teagasc, within the low-cost grass-based system to minimise concentrate use and maximise milk production from forage. The question was asked in terms of 'average' concentrate use over several years rather than in a given year. Some farmers answered in terms of concentrate given to cows per day at different times of the year, so different metrics are used in the table. Interview data were anonymised by giving the interviewees a number.

**Table 1.** Interviewee demographic information.

| Farmer Pseudonym | Location | Cow Numbers | Amount of Concentrate Fed per Cow | Position on Farm | Gender | Relationship to Other Interviewees |
|---|---|---|---|---|---|---|
| F1 | Northeast | 460 | 1500 kg/year | Owner | Male | n/a |
| F2 | Northeast | 200 | 1000 kg/year | Owner | Male | n/a |
| F3 | Midlands | 180 | 7 kg/day in winter, 2–5 kg/day in spring/summer | Owner | Male | n/a |
| F4 | Midlands | 130 | 500 kg/year | Owner | Male | n/a |
| F5 | Northeast | 400 | 600–700 kg/year | Owner | Male | n/a |
| F6 | Northeast | 260 | 1200 kg/year | Owner | Male | n/a |
| F7 | Southwest | 200 | 800 kg/year | Manager | Male | n/a |
| F8 | Southwest | 80 | No data | Owner | Male | n/a |
| F9 | Southwest | 170 | 650–700 kg/year | Owner | Male | n/a |
| F10 | Southwest | 200 | 800 kg/year | Owner | Male | n/a |
| F11 | Southwest | 250 | 2000 kg/year | Manager | Male | n/a |
| F12 | Southwest | 50 | 6 kg/day | Owner | Male | n/a |
| F13 | Southwest | 110 | 700 kg/year | Owner | Male | n/a |
| F14 | Southwest | 100 | 250–300 kg/year | Owner | Male | n/a |
| F15 | Southwest | 80 | 1700 kg/year | Owner | Male | n/a |
| F16 | Southwest | 80 | >1500 kg/year | Owner | Male | n/a |
| F17 | Southwest | 75 | 700 kg/year | Owner | Female | Partner of F20 |
| F18 | Southwest | 180 | 3 kg/year in summer | Owner | Male | n/a |
| F19 | Southwest | 130 | 500 kg/year | Owner | Male | n/a |
| F20 | Southwest | 75 | 700 kg/year | Owner | Male | Partner of F17 |

In the results section, the terms 'grass-based' or 'low-cost grass-based' are used to describe the system promoted by research and advisory bodies in Ireland of focusing on grass production as the engine of profitability and minimising concentrate use. The term 'higher-feed-input' describes a system where there is not a belief that concentrate use needs to minimised. Initial analysis was carried out of the notes taken after each interview. These notes were used to assess when data saturation was reached: when no substantially new content is generated from subsequent interviews [69].

Interviews were transcribed and analysed using Nvivo 12 software. Thematic analysis was carried out [70]. Thematic analysis involves reading each transcript and coding parts of the text into themes. These themes are then read for patterns that emerge within and across them. The analysis was part of a project looking at the role of grazing and year-round housing in dairy sectors in the UK and Ireland. The coding covered themes beyond those reported in this paper relating to structural and cultural factors hindering or fostering change towards meeting policy objectives in the Irish dairy sector. The research questions for this paper and the results presented were identified through an iterative inductive and deductive process consulting good farming literature on environmental norms and literature about grazing in the dairy sector presented in the introduction.

## 4. Results

### 4.1. Grass Production and Management as Good Farming

The messaging coming from research and advisory services in Ireland that maximising the grazed grass in the cows' diet is a way to ensure profitability and operate a relatively simple system was described in the introduction. The introduction showed some of the work that has been done to communicate this philosophy to farmers: Teagasc's dairy manual, farmer discussion groups and the development of grass management software. This section asks whether and how messaging about the value of grass production and management has been converted into good farming norms and identities.

The farmers interviewed described the skills, knowledge and facilities associated with grass management and the grass-based system as 'good farming':

> F7: 'Some farmers fall into high-input systems, because maybe they can't manage grass. With a low-input system, you need to have very high-quality grass. Some farmers are I suppose, refuse to be educated in grass measuring and that, that they just feed a lot of meal, and graze heavy covers during the summer, and cows milking well and they're happy. But it's non-profitable, it's not profitable.'

Here we see that in terms of the good farming theories of capital: 'high-quality grass', grass measurement and management skills and profitability are types of good farming cultural capital. A farmer would be judged by F7 as a 'good farmer' if they possess this type of cultural capital. According to this farmer, the issue is not that the practice of feeding more meal involves different types of cultural capital which belong to a different production system; rather, there is a deficit of cultural capital, which effectively makes a high-feed-input system unprofitable. Feeding more meal comes about because of a lack of investment by the farmer in the types of cultural capital associated with the grass-based system.

The use of grass was linked to profitability: several interviewees cited grass as a low-cost feedstuff:

> F4: 'Grass is the cheapest possible input, we'll say that you can have for cows. So, therefore it should be used to its max.'

This point about reducing costs as part of good farming is further explored in Section 5.1 below. An interviewee described how the grass-based system came to Ireland from New Zealand and changed the way farmers related to grass: it was treated as important feedstuff that needed to be carefully measured and managed rather than something that was taken for granted.

Interviewees spoke about how their grass management practices have changed as a result over time.

> F2: 'We measure grass. All the young farmers measure grass. I now know how to measure grass, you know, [laughs] we never did that before. [...] But now, the way they do it is so much better. Able to budget in front of them, knowing what you have in front of you for twenty-one days, whatever the cycle you're currently grazing in, and how to manage that. That's brilliant stuff.'

This farmer clearly values the grass management skills he and his team have acquired on their farm. Similarly, another farmer states that the principles of the grass-based system have been taught to farmers in the last 10–15 years:

> F4: 'And in Ireland, that rhetoric of trying to get cows out for longer, to grass and everything's going on for ten or fifteen year, and there's some people only changing now.'

This farmer's comment that 'there's some people only changing now' shows that he thinks these people are late adopters, behind the curve of good farming. Farmers defined their system in terms of a focus on grass:

> F5: 'It's pretty simple, we focus on growing grass as much if not more than actually on the cow itself.'

Interestingly, the farmers interviewed who identified as operating a higher-feed-input system emphasised the importance of grass and were not involved in separate social and information networks of farmers in the dominant grass-based system. A liquid milk producer states that he is not replacing grass with meal—he is supplementing it:

> Interviewer: 'And what made ye go down what you call the high-input sort of route in the Irish context, you know that's high input for Ireland?'

> F11: 'I suppose traditionally going back to the earlier discussion about the milk, we were always in winter or liquid milk, so we always had a high yielding cow; we always fed the cow well and looked after the cow as a priority. That's mainly it. Now I wouldn't feed excessively either to the extent that you're trying to replace forage with meal; you have to as I say back to profitability too, there's no point in feeding a cow out there and she walking out in the field and lying on lovely grass, either. So there has to be a balance.'

A farmer made the point that the metrics used in the grass-based system, low concentrate use and a long growing season, can be detached from the aim of profitability and become aims in their own right because they bestow status on farmers:

> F1: 'So, once you're performing and you're farming for profit not for milk, or for ego in the grass system, the ego, what I'm saying is having cows out on the first of February, [laughter] feeding them no meal, and having them out on Christmas Day; there's ego that way just as much as there's ego in the high-input system, to have ten or twelve thousand litre cows.'

This farmer's comments are wry, and he aims to point out the folly of farming for 'ego': aiming to build status according to the culture of the day rather than for profitability. The 'good farming' concept describes a similar mechanism at play: farmers aim to build status and succeed within the rules of their peer group. The research suggested that the principles of 'good grass farming' have successfully been established as the dominant culture of good farming among Irish dairy farmers because skills, knowledge and resources associated with the grass-based system are valued forms of cultural capital which bestow status on farmers.

*4.2. Intensive Grass Management as Good Farming*

The introduction showed a quotation from the Irish Government's 'Food Harvest 2020 report' which described Irish grazing systems as 'extensive': "Ireland's extensive, low-input grass-based production systems are the foundation of its green credentials [. . .]" [13] (p. 5). In the quotation, extensive production is associated with environmental sustainability, but the research showed that 'extensive' grass management, taken to mean minimising inputs and not aiming to maximise grass production per hectare, is not considered 'good farming' in the Irish dairy sector. Rather, the interviewees suggested that a more intensive, productivist system of using inputs to maximise grass production per hectare within certain parameters was considered good farming. In addition to skills and knowledge about *how* to manage grass in dairy farming, the volume of grass produced was also considered part of 'good farming'. Interviewees described their aim of producing more grass to produce more milk from the cows, which increases profits.

A farmer describes the change in norms that took place when Ireland entered into a grass-based system:

> F8: 'So now even, it's still going back to yield because people are still . . . I won't say blowing, but about their yields of grass. So, it's gone from yields of milk or yields of beef or whatever, to yields of grass. And people make plenty noise about how much grass they're growing now.'

Telling others about the volume of grass you produce is a way to displaying cultural capital and winning status. The rules of the game still create 'productivist' values but now focusing on volumes of grass rather than milk. According to this account, it is not the case that a grass-based dairy system can be considered 'extensive' and higher-feed-input one

'intensive'; rather, but both systems have the aim of maximising production, of grass or milk, respectively:

> Interviewer: 'And in your view what's a good dairy farmer?'
>
> F10: '[...] you can say performance wise, they must be hitting so many cows per hectare or so much milk solids per hectare, you know. But like look, they obviously have to be hitting within certain norms.'

Intensive, highly stocked dairy farms require fertiliser inputs to produce grass and in turn produce high volumes of manure per unit of land, both of which contribute to greenhouse gas emissions, ammonia emissions and water quality problems. A farmer who had some criticisms of Ireland's grass-based system on environmental grounds states that not maximising grass production using inputs was considered a 'low achievement':

> F8: 'Now, we've all gone, conventional agriculture has gone completely natural to an automatic high-input system for forage structure. It's not considered on a low-input basis because it's seen as a waste of resources or as low achievement.'

By an 'automatic high-input system', the interviewee means using fertiliser to promote forage growth. This links use of inputs to good farming: not using inputs to maximise grass production from the land is not good farming. Interestingly, while the dominant story told in Ireland that a higher-feed-input system is expensive, inefficient and dependent on inputs which increased farmer's running costs and capital costs, an intensive grass-based system using fertiliser to produce more grass was considered efficient 'good farming'. A system involving maximising fertiliser inputs within legal parameters is seen by many interviewees as 'good farming', but it is also criticised on environmental grounds [64]. Government targets to reduce nutrient pollution and nitrous oxide emissions that involve a reduction in fertiliser use could conflict with current 'good farming' practice in the Irish dairy sector.

*4.3. Potential for Low-Input Grass Management as Good Farming*

This section will look at how the use of multispecies swards fits with current conceptions of 'good farming'. Farmer interviewees were asked about their views on the potential of multispecies swards including clover to lower the fertiliser requirement of grass production on Irish dairy farms. The majority of dairy farmer respondents stated that they were interested in trying mixed swards, or a few had tried them, because they knew mixed swards could lower fertiliser use. A few interviewees also expressed misgivings about the amount of fertiliser that was currently used to drive grass production:

> F19: 'I can see huge potential for it [clover], huge potential for it. Obviously, the clover, it pulls the nitrogen from the atmosphere into the soil, and if it stops you spreading artificial fertiliser, sure there's a huge benefit in it.'
>
> F11: 'We need to do a bit more reseeding and maybe incorporating clover or the mixed herbal leys or something; just try and get more with less fertiliser. Even just outside of that [the cost of fertiliser] like environmental as well like you know. There's more organic ways of hopefully growing grass than having to be pumping a load of chemicals in too like.'

As well as the environmental benefits of reducing fertiliser use, farmer interviewees spoke about the production benefits of clover and mixed swards: milk production can be maintained or even increased with less fertiliser application. Farmers' desires to farm in more environmentally friendly ways can be built on in policy and advisory initiatives, as expanded on below in Section 5.3.

Multispecies swards were described by interviewees as more challenging to manage than perennial rye grass, and interviewees felt that they currently lacked skills and knowledge. The management challenges were seen to stem from different grass species being suited to different types of ground, the potential of clover to cause bloat in cattle and

different growth rates of clover and perennial rye grass at different times of year. Farmers describe the complications of managing clover on their farm:

> F7: 'There's three different types of ground, I suppose, in the one block; further down it's a very wet land, I suppose, clover mightn't survive in it. Here on the dairy block, it's dry; clover will survive. And then over in the middle there's both. So, I suppose the one thing we can't have is three different types of grazing mixes, because it's harder to manage. If there's cows going from just grass only to clover, you have problems with bloat, because cows will gorge on clover. You'd have to start putting up twelve-hour wires, and it's just harder management.'

The interviews suggested there was appetite for more environmentally friendly grass production practices, but there was a need for help developing skills and knowledge to make management possible. If there is a desire to promote grass management practices that involve less fertiliser use, the skills and knowledge needed to manage multispecies swards could in time be a higher value form of cultural capital than the skills and knowledge needed to produce perennial rye grass using fertiliser inputs to drive production. This point is expanded on below in Section 5.2. An environmentally conscious farmer reframes the use of fertiliser from a vital part of the grass-based system to a disruptive addition to natural nutrient cycling processes:

> F8: 'And now obviously without getting into the microbiology, in any system if you add something in biology, something else goes ... if you add too much sugar to your diet it affects your insulin system, and you eventually can become diabetic because your body has stopped producing. Similarly, if you add a lot of nitrous to the cycle, the nitrogen cycle in the soil is interrupted and maybe made a bit redundant and diminished and whatever. So, there is an actual inhibiting quality to the nitrogen to the output, or to the functionality of the diversity of the multispecies forms.'

Here, he frames the use of fertiliser as not part of good farming because it diminishes the potential of the soil to cycle nitrogen naturally. Just as farmer 7 describes a higher-feed-input system as a system that a farmer might fall into it because of a lack of skills and knowledge to manage grass, here also, the use of nitrogen fertiliser is reframed from an essential part of the grass-based system to a disrupting and inhibiting input. He identified interest in multispecies swards as 'good farming':

> F8: 'And the now most progressive ... I won't say, progressive is the wrong word, but the most aware or engaged dairy farmers are aware of multispecies swards by, I would say, an inordinate magnitude than they were twelve months ago.'

This farmer describes 'good farming' as engagement with multispecies swards, which shows a desire for change is in his definition of good farming.

## 5. Discussion

The results showed that the grass-based system in Ireland of maximizing milk from grass and minimising the use of bought in feed had successfully become identified with 'good farming', according to the farmers interviewed in this study. The skills and knowledge in grass management associated with the grass-based system were valuable forms of cultural capital. In relation to the environmental impacts of the grass-based system, while it was framed in a government report as 'extensive', these results suggest that 'extensive' production within the grass-based system would mean minimizing fertiliser use and having a low stocking rate, and these practices were not part of the dominant definition of good farming. The dominant definition of good grass-based farming was productivist and intensive: aiming to maximise grass and milk production from the land using fertiliser inputs, within legal parameters. Producing high volumes of grass could be taken as a demonstration of skill. Given that producing a high grass yield is dependent on using high volumes of fertiliser, government targets to reduce fertiliser use could conflict with current productivist, intensive good farming norms in the dairy sector. This paper analysed

views on the potential for the uptake of multispecies swards in grass management which could reduce fertiliser use. The results showed some appetite from farmers for reducing the environmental footprint of current grass management practices but perceptions of a deficit in skills and knowledge in how to implement multispecies swards.

The discussion will explore three areas in more detail: firstly, the potential to frame low fertiliser usage as good farming because it is cost saving; secondly, the potential for the skills needed to manage multispecies swards to be understood as good farming; and thirdly, the mechanisms by which these changes could come about.

### 5.1. Good Farming as Practices That Lower Costs

As described in Section 2, good farming ideals develop around production methods that are profitable in a given context [38]. The price of agricultural commodities and inputs, and the availability of agricultural subsidies, can influence whether 'good farming' means maximizing yields or reducing inputs and yields [9,22,30]. In the face of economic pressures, farmers may define 'good farming' as increasing production to achieve economies of scale or, alternatively, reducing inputs and lowering costs [38]. Research has shown that productivism is still important to farmers as a good farming ideal [40]. New-entrant Irish dairy farmer respondents to a survey rated maximizing production as the fifth-most important farming objective after objectives relating to profitability and quality produce [7]. According to farmer 8 quoted in Section 4.2, there had been a change in productivist values in the Irish dairy sector: yields of milk were replaced with yields of grass as a valued form of cultural capital in the grass-based system.

The development of the grass-based system is linked to profitability. Using feed inputs to increase milk yields is framed by research bodies in Ireland as high cost and uneconomical, whereas under current prices, increasing grass yields through application of fertiliser is seen as economical [43,46]. Running a profitable farm has been shown in other research to be part of a definition of 'good farming' [22,31,32]. The results here showed that interviewees linked good grass-based farming with profitability.

Messaging about the cost of inputs has been shown in other studies to trump productivist good farming ideals. Huttunen and Peltomaa [71] showed how policies aimed at encouraging Finnish farmers to use less fertiliser and optimize the timing and method of fertiliser application did succeed in creating good farming ideals around these practices. The policies inspired farmer to invest in knowledge and facilities need for reduced fertiliser application: 'The changes in the policy have affected the fertilisation practices in ways that have helped in internalising the objectives behind the reductions.' (p. 222). The government has set out a target to lower fertiliser use on Irish farms [61]. In order to achieve this policy target, advisory services could frame the economics of using fertiliser not only in terms of the current price of fertiliser but with respect to locking in to a system dependent on inputs with fluctuating prices, and the externalized costs of production, even if it means producing less. Given that the Russia-Ukraine conflict has resulted in an increase in fertiliser prices internationally [72], this finding is relevant to dairy sectors in other countries which are also highly dependent on fertiliser inputs.

### 5.2. Good Farming Skills and Knowledge

This research explored whether the philosophy of the grass-based system of minimizing concentrate feed and focusing on milk production from grass was translated into a commonly held definition of good farming among Irish dairy farmers. Skills and knowledge are forms of embodied cultural capital [73]. Given that managing multispecies swards requires new types of knowledge and skill sets compared with current best practice grass management, there is potential for these practices which lower fertiliser use to be identified as 'good farming' in the Irish dairy sector. Burton et al. (2008) state that there are three conditions for a practice to become identified as 'good farming': the practice must be culturally relevant to farmers; there must be observable, visible results from the practice; and these results must be accessible to other members of the farming community.

In relation to the first criterion of cultural relevance, there is also research showing that environmental sustainability in and of itself is now a culturally relevant norm to many farmers, particularly if they have taken part in environmental schemes or changed to environmentally friendly forms of production [22,30,32]. Farmers in England recognized the skills needed to produce grass using less fertiliser as a valuable form of cultural capital on environmental grounds [30]. This was echoed by several interviewees' comments in Section 4.3 that they wanted to adopt grass management practices that involved less fertiliser use for environmental reasons. This adds to the body of international examples of farmers being aware of and concerned about their environmental impacts.

In addition practices which link production and environmental sustainability such as conservation tillage practices can be considered culturally relevant because they are still production focused but have the potential to bring about environmentally sustainable outcomes [74]. The same could be true of multispecies swards, which need not involve a reduction in grass production, but involve a reduction in the use of environmentally detrimental inputs.

The last two of Burton et al. (2008) criteria for a practice to be established as good farming are a visible demonstration of skill and that they must be accessible to other farmers, which can also be called 'roadside farming' [9]. A description of 'roadside farming' can be seen in the quote in Section 4.1 from farmer 1 that within the grass-based system farmers will try to have their cows out on Christmas day or the first day of February to farm for 'ego', i.e., gain status within the grass-focused 'rules of the game' when other farmers see this practice. Research has shown that certain practices lend themselves to visible demonstration better than others. For instance it could be difficult to include management of waterways in a definition of good farming because water course management may not produce visible symbols of good farming [39]. In contrast, conservation areas on farm can be valued by the farmer for the wildlife they produce [22,30,31] and displayed to visitors as indications of the farmers' commitment to sustainability [30,32] and taken as a sign of skill that a farmer can cultivate wildlife while producing food [32]. The Irish government has set out policy to increase the uptake of multispecies swards [61]. In order to meet this policy objective, visible symbols associated with successful cultivation of multispecies swards could be identified and used in knowledge exchange. Just as some interviewees saw practices associated with feeding more concentrate as reflecting a deficit in skills and knowledge, compared to the skills and knowledge needed in the grass-based system, the same could become true of practices of using multispecies swards which lower fertiliser use. While, as described in the background section, Ireland has a unique advisory landscape that allows for efficient exchange of consistent messaging, this finding about the value of cultural relevance and visible symbols of mixed sward cultivation are also relevant to international contexts with more diverse knowledge exchange landscapes [49].

*5.3. Translation of Good Farming Ideals into Practice*

The way in which a definition of good farming came to be understood as good grass management in the Irish dairy sector accords with literature on how good farming symbols and identities are produced.

As we saw in Section 2 good farming can be shaped by the development of technology [9]. Different technologies are integral to the operation of the grass-based system which are required for grass growth, grass measurement, grass budgeting, etc. [75]. Good farming ideals around the use of certain technologies do not always translate into practice however [31,42]. For instance McDonald et al. (2016) found that 80% of new entrant dairy farmers surveyed recognized grass budgeting as important to their farming needs, but 51% implemented it. Other factors can mean that practices which are understood to be beneficial are not implemented, such as time commitment and need to develop new skills [5]. Thus, in relation to multispecies swards, as the interviewees described, the ease of use and management influence their decisions about whether or not to use them, in addition to how multispecies swards fit with good farming identities and ideals. The government industry

report Food Vision 2030 called for more research on 'Grass, herbs and fodder varieties that deliver required sward yields and longevity at lower levels of nitrogen application.' [61] (p. 59). This could include research on farmer priorities in the usability of multispecies swards.

In relation to the influence of advisors and policy, as described in Section 1, there are a number of policy documents and initiatives aimed at bringing about sustainable dairy production practices [61,62,76]. Peer-to-peer deliberation and translation of extension advice are valuable ways to ensure that information becomes culturally relevant and understood in the context of farmers' individual circumstances [75]. A study on new-entrant dairy farmers concluded that the more equal power dynamics are in a knowledge exchange relationship, the more potential there is for actors to meaningfully influence each other [77]. The good farmer concept is underpinned by questions about power: who has the power to define what a 'good farmer' is and for what purpose. 'Good farming' can be understand as prescriptions about how agriculture should be carried out by experts, or as a form of cultural resistance to change by farmers [9]. This could be borne in mind in the context of the future development of grass-based good farming ideals in allowing farmers to co-determine what 'good farming' is in the Irish context in tandem with scientists, industry stakeholders and farm advisers. The comments by farmer interviewees in Section 4.3 that they would like to see lower-input grass-management practices adopted could be a basis for shaping future grass-based good farming ideals. In terms of co-producing outcomes, the Irish government carried out public dialogue events ahead of the publication of the agri-food strategy 2030 [78], and the Signpost Programme involves a number of dairy demonstration farms to facilitate peer-to-peer learning on the environment [76]. Another option to include different perspectives in policy making is to commission a farmers' group to carry out work on how to achieve environmental targets, similar to an initiative in Scotland [79]. These results have relevance to an international context because they provide greater understanding of farmer involvement in a grass-based system, as called for in previous research [1].

Qualitative interviews are a valuable method for gaining in-depth understanding of the factors that shape farmer decision making [20]. There are limitations to this method, however, as the in-depth nature of qualitative research limits the number of participants involved in the study. Purposive sampling was used to access interviewees with a range of perspectives from different areas, ages, production systems and values. Women were under-represented because participants were recruited from a survey which received the majority of responses from male farmers. The gendered aspects of farming are another area which has not yet been explored in depth using the good farming concept [9]. The majority of interviews took place in the southwest of Ireland. Including more female interviewees and interviewees in other areas of Ireland may have yielded different or additional insights to the ones identified in this paper. However, these results can be taken to indicate the prevailing attitudes towards good farming norms in Ireland and as providing insights for fostering sustainable production practices in other contexts.

## 6. Conclusions

Today, the environmental benefits of different ruminant production systems are debated, with claims about the environmental benefits of grass-based production over higher-feed-input systems [1,57]. This paper adds nuance to the discussion about environmental impacts by drawing attention to different production practices within the grass-based system. As the result of concerted industry and advisory effort, good farming norms in Ireland moved away from productivism focused on output of milk to productivism focused on outputs of grass. The research showed some appetite among farmers to reduce fertiliser use in grass production which could be built on to meet environmental policy aims. The paper identified three strategies for facilitating a definition of good farming which involves low fertiliser use: emphasising the cost-saving aspect of reducing fertiliser; identifying visible symbols of 'good farming' using multispecies swards; and co-producing the definition of

good farming with a diverse range of stakeholders including farmers. These results could be extended to other countries where there is an aspiration to foster a transition to more sustainable grass-based livestock production systems.

**Funding:** This work was funded by a British Academy postdoctoral fellowship (London, UK, grant number pf170071). The work received additional funding from the Scottish Government Rural and Environment Science and Analytical Services Division, as part of the Centre of Expertise on Animal Disease Outbreaks (EPIC; Edinburgh, Scotland).

**Institutional Review Board Statement:** Ethical approval was obtained from the James Hutton Ethics Committee application number 179/2019.

**Informed Consent Statement:** Informed consent was obtained from all subjects involved in the study.

**Data Availability Statement:** Data are not available because confidentiality and anonymity were guaranteed to study participants.

**Acknowledgments:** The author would like to thank the farmers who gave up their time to be interviewed for the research.

**Conflicts of Interest:** The author declares no conflict of interest.

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
