# Peer review of "A Qualitative Study of Irish Dairy Farmer Values Relating to Sustainable Grass-Based Production Practices Using the Concept of ‘Good Farming’"

_sustainability, doi:10.3390/su14116604_

Round 1

Reviewer 1 Report

I thank the authors the their revision of their manuscript. I appreciate the changes and have only few additional comments:

  • The quote from lines 174-182 feels a bit too long. I suggest using an indirect quote here, and focus on the core arguments of the EPA.
  • Section 3: Please add the period over which the data was collected (and ideally when the survey, through which the farmers were recruited, was conducted).
  • The sub-heading numbering in section 4 is erroneous.
  • l.538: is there a "they" missing? (...and that THEY must be accessible)
  • The indentation of the quotes did not "survived" the manuscript formatting. In this case, I suggest to use quotation marks around them.

Author Response

Reviewer 1

I thank the authors the their revision of their manuscript. I appreciate the changes and have only few additional comments:
Thank you for your positive comments.

  • The quote from lines 174-182 feels a bit too long. I suggest using an indirect quote here, and focus on the core arguments of the EPA.
    I took out the quote and summarised the main messages.
  • Section 3: Please add the period over which the data was collected (and ideally when the survey, through which the farmers were recruited, was conducted).
    These details were included.
  • The sub-heading numbering in section 4 is erroneous.
    Thanks for spotting this. The numbers on the discussion and conclusion were changed.
  • l.538: is there a "they" missing? (...and that THEY must be accessible)
    This was corrected.
  • The indentation of the quotes did not "survived" the manuscript formatting. In this case, I suggest to use quotation marks around them.
    I put quotation marks around the quotes.

Reviewer 2 Report

The paper “A qualitative study of Irish dairy farmer values relating to sustainable grass-based production practices using the concept of ‘good farming’” presents interesting and important findings about the attitude of the Irish farmers towards the grass-based dairy systems. The introduction and the background sections well describe the theoretical and past, current, and future socio-economic and public policies challenges of the dairy sector in Ireland and in the EU.

There are some issues that can improve the overall paper quality:

- the methodological framework of qualitative interviews and the main policies implications of the main results are not clearly described in the abstract.

- material and methods – the author should present other papers using the same methodology for data collection to explain the relevance of the 20 qualitative interviews. The distribution of the farmers in the country is not well justified. Additional details about data transcript management and advantages and disadvantages of the qualitative research methodology will improve the overall paper quality.

-  the policy implications in the discussion section can be improved.

Author Response

Reviewer 2

The paper “A qualitative study of Irish dairy farmer values relating to sustainable grass-based production practices using the concept of ‘good farming’” presents interesting and important findings about the attitude of the Irish farmers towards the grass-based dairy systems. The introduction and the background sections well describe the theoretical and past, current, and future socio-economic and public policies challenges of the dairy sector in Ireland and in the EU.

There are some issues that can improve the overall paper quality:

- the methodological framework of qualitative interviews and the main policies implications of the main results are not clearly described in the abstract.
Qualitative interviews are mentioned on line 15 of the abstract. The paper includes policy implications on lines 27-32 of the abstract.

- material and methods – the author should present other papers using the same methodology for data collection to explain the relevance of the 20 qualitative interviews.
Other papers that carried out a similar number of interviews were included and a note about data saturation in the interviews was added to clarify why the number of interviews was chosen on lines 273-280.

The distribution of the farmers in the country is not well justified.
I included additional information about why most interviews were carried out in the south west on lines 271-273.

Additional details about data transcript management and advantages and disadvantages of the qualitative research methodology will improve the overall paper quality.
Details of the management of the data and transcription were included on lines 290-292.

-  the policy implications in the discussion section can be improved.
I added to the policy implications in the discussion section by adding more detail of which policies the work is relevant to and suggestions for meeting policy goals.

Reviewer 3 Report

The paper is well written, grounded on sound methods and definitely interesting. However, I have some suggestions to improve its readability, logical flow and structure. Additionally, I have a few minor remarks.

- Abstract: I suggest to try to better clarify how the paper is "using" the good farmer framework. That sentence (L14) is crucial and sounds a bit confusing to me. I also suggest to give a clue of what a multispecies sward is already in the abstract (here there is a reference to fertilizer use, so maybe specifying that they include legumes is sufficient)

- Introduction: definitely well written, but it has somehow the same problem I found in the abstract. How the study is making use of the "good farming" framework? The introduction goes from a detailed description of the good farming definition (until L106) to the research question, but in my opinion a link is missing. Should there be maybe something about how grazing is considered in the different definitions of good farming over the years? Or something about how this aspect is neglected and this study is needed? Or both? Could the subtitles of the Results section be of any help? They summarize what the study is actually studying through the interviews, maybe the introduction could refer to at least some of these topics/concepts

- Background: necessary, but too extensive, and the logical connection with the introduction is not cristal clear. One would expect Methods and Results after the research question at the end of the Introduction, but I see that this is not a typical "hard science" paper. For instance: the whole text helps to give context, but especially from L183 to L212 the description gets way too detailed and the focus of the paper is a bit lost, in my opinion. I propose to shorten this section and to try to relate it to the theoretical framework and the research question of the Introduction. Maybe it´s the other way round: some clues of the background should be given in the Introduction (how the sector evolved, and how this could have impacted the "good farming" definition over time? Are the failed environmental objectives pushing towards a more comprehensive definition? How the study comes into play in this?). Finding connections between the research question and the background would improve readability, conciseness and logical flow at once. I highly suggest such an effort.

- Results/Conclusion. The outcomes highlighted at L619-621 seem poorly reflected in the Results section, or at least it is not so easy to understand to what in the Results the author is referring. I suggest to use all possible ways allowed by the journal guidelines to highglight relevant findings in the Results section (text in italics?).

Minor remarks are the following: 

- L41-43, sentence starting by "In addition": it seems a little bit out of context and very generic. Maybe better as introductory sentence of the following subsection (L48)?

- L45, Please check, a sentence seems to be broken 

- L53, "grass" is fine if you think that grasslands are made by just one grass species (which in most cases is correct), but we are talking about multispecies swards. In my opinion, "grasses" would make the sentence generally more valid. There could be more than one grass species in a multispecies sward.

- L46, chicory is a good example of a herb, I can suggest another one: plantains. It should be possible to find some good Irish literature about the introduction of plantains for environmental goals in multispecies swards 

- L80, I suggest to put "norm" within quotes

- L84, "symbols". Would a definition or an example of such symbols help the reader to better understand? They seem to be important, they are mentioned also in the Conclusions

- L167-170, since carbon sequestration in grassland is quite debated and debatable, I suggest to include also a less optimistic view in the reasoning (for instance: what if C in the soil is already plateauing?). It should be possible to find supporting and unsupporting literature.

- L190, livestock manure nitrogen per hectare

- Discussion is section number 5, the following sections and subsections have to be renumbered

Author Response

Reviewer 3

The paper is well written, grounded on sound methods and definitely interesting. However, I have some suggestions to improve its readability, logical flow and structure. Additionally, I have a few minor remarks.
Thanks for your positive comments and constructive feedback.

- Abstract: I suggest to try to better clarify how the paper is "using" the good farmer framework. That sentence (L14) is crucial and sounds a bit confusing to me.
I changed the wording of the first research question in the abstract, hopefully it’s clearer now.

I also suggest to give a clue of what a multispecies sward is already in the abstract (here there is a reference to fertilizer use, so maybe specifying that they include legumes is sufficient)
I included two references to legumes in the abstract.

- Introduction: definitely well written, but it has somehow the same problem I found in the abstract. How the study is making use of the "good farming" framework? The introduction goes from a detailed description of the good farming definition (until L106) to the research question, but in my opinion a link is missing. Should there be maybe something about how grazing is considered in the different definitions of good farming over the years? Or something about how this aspect is neglected and this study is needed? Or both? Could the subtitles of the Results section be of any help? They summarize what the study is actually studying through the interviews, maybe the introduction could refer to at least some of these topics/concepts.
Thanks for the comment, I had a bit of trouble understanding what you mean. I added information on lines 123-130 which hopefully address this comment.

- Background: necessary, but too extensive, and the logical connection with the introduction is not cristal clear. One would expect Methods and Results after the research question at the end of the Introduction, but I see that this is not a typical "hard science" paper. For instance: the whole text helps to give context, but especially from L183 to L212 the description gets way too detailed and the focus of the paper is a bit lost, in my opinion. I propose to shorten this section and to try to relate it to the theoretical framework and the research question of the Introduction. Maybe it´s the other way round: some clues of the background should be given in the Introduction (how the sector evolved, and how this could have impacted the "good farming" definition over time? Are the failed environmental objectives pushing towards a more comprehensive definition? How the study comes into play in this?). Finding connections between the research question and the background would improve readability, conciseness and logical flow at once. I highly suggest such an effort.
Thanks for this point. This section was originally part of the introduction and another reviewer suggested putting it as a separate section, which might explain why it now doesn’t flow that well. I added some sentences into the introduction on lines 59-62 about the environmental challenges in Ireland to link to the background section. I put “signposting” sentences at the beginning and end of the background section.

I see your point that the section is quite long and there is some redundancy for communicating the points that the Irish sector is very focused on grass and is also facing environmental challenges. I reduced some of the detail. I think my original reasoning was that the environment is a controversial and difficult subject in the Irish dairy sector at the moment, so I wanted to make the premise of the paper that there is a need for environmental action ironclad, by listing all the problems and all the policy objectives and instruments to improve environmental outcomes. I’d like to keep the rest of the detail in for an Irish audience for that reason.

- Results/Conclusion. The outcomes highlighted at L619-621 seem poorly reflected in the Results section, or at least it is not so easy to understand to what in the Results the author is referring. I suggest to use all possible ways allowed by the journal guidelines to highglight relevant findings in the Results section (text in italics?).
I wasn’t sure where you meant I should put the italics. The three points in the sentence L619-621 correspond to a section in the discussion and I put in a few sentences in the results section that signpost the connection with these sections.

Minor remarks are the following: 

- L41-43, sentence starting by "In addition": it seems a little bit out of context and very generic. Maybe better as introductory sentence of the following subsection (L48)?
Thanks for this comment. I didn’t see the problem with this sentence and the flow of the paragraph though. I’d prefer to keep it as it is if that’s okay with you and the editor.

- L45, Please check, a sentence seems to be broken
This was corrected.  

- L53, "grass" is fine if you think that grasslands are made by just one grass species (which in most cases is correct), but we are talking about multispecies swards. In my opinion, "grasses" would make the sentence generally more valid. There could be more than one grass species in a multispecies sward.
This was changed.

- L46, chicory is a good example of a herb, I can suggest another one: plantains. It should be possible to find some good Irish literature about the introduction of plantains for environmental goals in multispecies swards 
A reference for plantain was also included.

- L80, I suggest to put "norm" within quotes
Done.  

- L84, "symbols". Would a definition or an example of such symbols help the reader to better understand? They seem to be important, they are mentioned also in the Conclusions
Examples were included on line 100.

- L167-170, since carbon sequestration in grassland is quite debated and debatable, I suggest to include also a less optimistic view in the reasoning (for instance: what if C in the soil is already plateauing?). It should be possible to find supporting and unsupporting literature.
A reference was included.

- L190, livestock manure nitrogen per hectare
Thanks for spotting this, this was corrected.

- Discussion is section number 5, the following sections and subsections have to be renumbered
This was corrected.

This manuscript is a resubmission of an earlier submission. The following is a list of the peer review reports and author responses from that submission.

Round 1

Reviewer 1 Report

This manuscript is interesting to read and well written to be thinkable topic.

I would like to point out the following several issues.

Title,

- The title of this paper started with a question, but it seems difficult to draw a clear conclusion to that question. How about editing the title?

Introduction,

- Line 188 : Various concepts of ‘good farming’ were mentioned in detail, and then could you describe what your definition for the good farming? In addition, is ‘good farming’ a general and academic phrase?

- Line 197 : Why was it mentioned as a ‘qualitative interview’?

Materials and methods,

- Line 205~207 : How about replacing ‘small numbers’ with percentage(%) because ‘detailed information’ and ‘small number’ could be misunderstood?

- Line 243 : No title on table 1.

Results,

- Line 299 : ‘ye’ typo?

- Line 369-370 : ‘fertiliser’ is a way of conventional agriculture? Could you explain in detail for the phrase (…using fertilizer… good farming) in line 370?

- Line 375 : Start with 3.2.

- Line 434 : ‘bad farming’ is the just opposite meaning of good farming?

Discussion & conclusion,

- Line 493 : ‘p.222’ typo?

- Detailed discussion and conclusion were well described. I would suggest that you could combine the discussion with results and make a separate section for a conclusion. Then it seems to be able to deliver much clear contents.

Reviewer 2 Report

I had the pleasure to read and review the paper “Is grass-based dairy farming ‘good farming’? A qualitative study of Irish dairy farmer values relating to sustainable grass-based production practices”. Using qualitative interviews, the authors study farmers’ attitudes towards different aspects of pasture-based dairy productions and root their approach in the “good farming”-framework. I believe the understanding farmers views and attitudes from the perspective of “good farming” is an important contribution to facilitate changes in farming system and that the paper can be of interest for a wide readership. Nevertheless, I think that the paper should be revised before publication. In the following, I list a number of major and minor points the authors may consider:

Major points:

  • The paper’s strong emphasis on the Irish background (especially in the introduction), does not help argument in l. 578, that the results are relevant for an international context. The detailed background is definitely helpful to understand the current situation in Ireland, but I think it takes to much room in the introduction. I suggest moving this material (particularly l. 49-141) to a background section (or alternatively an appendix) and focusing the more “general” issues in the introduction.
  • This would also shorten the introduction, which is quite long.
  • The definition of norms seems to be quite relevant, I would include (and potentially extend) footnote 1 in the main text.
  • The presentation of the applied methodology is insufficient.
    • How structured were the interviews?
    • Some additional information (e.g. lengths of the interviews) could be given.
    • It would be helpful for readers not engaged in qualitative research to give more information about the chosen analysis methods.
    • The structure of section 3 suggests that the research questions were used as overall themes (and thus identified before the analysis), is this correct?
    • Related, a motivation why the specific method was chosen is missing.
  • The results section is very hard to read, because it is difficult to differentiate quotes from the farmers and the actual text.
  • In addition, the section relies heavily on quotes (over 50 % in section 3.3.). Is it possible to reduce this, e.g. by shortening and including some of them in the running text?
  • The discussion could benefit from a short reflection of the chosen methodology.
  • The relevance to other contexts, or to which extent the findings may be used in other contexts, could be discussed in more detail.
  • Given the length of Section 4, the conclusions could be moved to a new section 5.

Minor points:

  • L. 11-13: I suggest splitting the sentence
  • L. 18: “a concerted efforts”: should be “effort”?
  • There are multiple instances where the citation is not consistent (e.g. l. 185, 560 and 567)
  • The subsection numbering in section 3 is incorrect.
  • L. 82-83: how is 40% “largely” a doubling
  • L. 221-224 feel like a part of the general discussion
  • L. 229: which advise exactly?